# Empowering and Building the Capabilities of Mid-Level Health Service Managers to Lead and Support the Health Workforce—A Study Protocol

**DOI:** 10.3390/ijerph21080994

**Published:** 2024-07-29

**Authors:** Zhanming Liang, Jemma C. King, Cate Nagle, Tilley Pain, Andrew J. Mallett

**Affiliations:** 1College of Public Health, Medical and Veterinary Sciences, James Cook University, Townsville 4810, Australia; jemma.king@jcu.edu.au (J.C.K.); tilley.pain@health.qld.gov.au (T.P.); 2College of Healthcare Sciences, James Cook University, Townsville 4810, Australia; cate.nagle@jcu.edu.au; 3Townsville Hospital and Health Service, Townsville 4810, Australia; andrew.mallett@health.qld.gov.au; 4College of Medicine and Dentistry, James Cook University, Townsville 4810, Australia; 5Institute for Molecular Bioscience, The University of Queensland, St Lucia 4067, Australia

**Keywords:** health managers, management competency, management capacity, clinical staff, burnout, retention

## Abstract

(1) Background: Mid-level managers in healthcare are central to improving safety and quality of care. Their ability in demonstrating leadership and management competency in their roles and supporting frontline managers and frontline staff has a direct effect on staff retention and turn-over. Yet, investment in their professional development and support for mid-level managers is often neither adequate nor effective, and high rates of staff turnover are evident. This study, set in northern Queensland, Australia, takes a strength-based approach to explore the role and strengths of mid-level managers and organisations’ existing mechanisms in supporting managers. With broad involvement and contribution from managers at different management level and frontline staff, the project will identify strategies to address the challenges mid-level managers face while building on their capabilities. (2) Methods: Using co-design principles, a situation analysis approach will guide a mixed-methods, multiphase design. Qualitative data will be collected using transcripts of focus groups and quantitative data will be collected by surveys that include validated scales. (3) Results: Thematic analysis of the transcripts will be guided by the framework of Braun and Clarke. Quantitative data will employ descriptive and inferential analysis, including chi-squared, *t*-tests, and univariate analyses of variance. (4) Conclusions: This study will generate evidence to guide two partner organisations, and other similar organisations, to develop strategies to improve support for mid-level managers and build their capabilities to support and lead frontline managers and staff. Competent mid-level managers are critical to high-quality patient care and improve the outcomes of the population they serve.

## 1. Introduction

In healthcare, middle-level (hereafter, mid-level) managers play a key role in bridging senior executives and frontline managers/staff. They are responsible for organisations’ operational matters and are instrumental in improving quality patient care [1,2] and creating a supportive work environment that can have a positive effect on staff retention [3,4]. In the rapidly changing healthcare climate, healthcare organisations have been pressured to innovate, including the adoption of digital technology. Mid-level managers are expected to have the ability to lead and manage changes and transition, such as implementing an innovative practice that facilitates positive chance in the organisation and quality improvement initiatives that improve patient experience of care. However, mid-level managers with mixed clinical and management responsibilities often lack opportunities to develop leadership capabilities, which may impact their self-confidence and job satisfaction [5,6]. The resultant high turnover amongst mid-level managers has implications for the directly impacted staff and cascading effects upon the organisation, which may manifest in a reduced quality of patient care and staff productivity [6]. Therefore, developing a better understanding of how to best support and build the capabilities of mid-level managers in fulfilling their roles has strategic importance for healthcare organisations.

Mid-level management is defined by the reporting structure in healthcare originations as detailed in Liang et al.’s management competency study in health service managers [7]. With the Chief Executive of the organisation holding the position at the first management level, based on the reporting structure, mid-level managers are those at the third or fourth management level depending on the size and complexity of the management structure of the organisation. For the purposes of this study set in northern Queensland, Australia, mid-level managers are defined as managers that hold a Band 6 or 7 management position in nursing, allied health/health promotion, and medical services similar to level 3 and level 4 managers based on the above classification.

It is well established that continued professional development opportunities are essential for individual development, staff retention, and health system capacity building [8]. This equally applies to mid-level managers who must have the capability to oversee their important management functions and fulfil their management roles, potentially necessitating additional development opportunities. Evidence has demonstrated that improved management competency can lead to improved management and service outcomes [9,10]. Hence, investment in supporting mid-level managers and developing their leadership and management capabilities is a worthwhile investment [11,12,13]. To achieve sustained improvement, continually building management capacity at the organisation and system levels is essential. In addition, developing the organisation’s internal mechanisms to enable mid-level managers to demonstrate their leadership and management qualities within their roles is also critical. Developing the competence of mid-level managers to better support and enable frontline managers and other staff to continue to provide responsive care within their full scope of practice is one mechanism to ensure the sustainability of quality care provision [14,15,16].

It is important for organisations to recognise managers’ important roles within the work and organisational environments and the broader health system [16]. It is also critical to acknowledge factors in the work environment beyond the managers’ control restricting their capacity, which must be addressed. Providing mid-level managers the opportunity to identify support that is important to them and prioritising strategies in addressing these factors are necessary. This contributes to developing mid-level managers’ understanding of their own strengths, empowering them in committing to professional growth, and building their confidence, ability, and agency to undertake the leadership and management aspects of their roles. In addition, staff empowerment has proven to contribute to improved staff job satisfaction and retention [17]. The factors that influence decisions to leave or remain in any position are multifactorial [18], but this can be further compounded when working in rural and remote areas where there are noted to be “critical workforce maldistributions” [19]. In the face of the aging health workforce and the misdistribution of the workforce in regional and rural areas, efforts to attract and retain highly capable staff in rural and remote areas are needed to sustain health services’ capability in addressing the healthcare needs of the communities they serve [20,21].

The above understandings are captured in a proposed conceptual framework in Figure 1, demonstrating a process to further develop the mid-level management workforce with the capacity to lead and support frontline staff.

In Queensland, one of the six Australian States, Hospital and Health Services (HHSs) are the statutory bodies responsible for providing public health services in Queensland using service agreements to determine the scope of services provided and the funding received for the provision of the services from the Queensland Department of Health [13,22]. Five HHSs the population in North Queensland—the decentralised regional and remote areas of Queensland. These five HHSs include Torres and Cape; Cairns and Hinterland; North West; Townsville; and Mackay [23,24]. Each of these HHSs represents a different but expansive geographical area, requiring that healthcare provision caters to the different demographic, social, and health needs within the service boundary. The urgent needs of addressing the “critical workforce maldistribution” present in rural and remote Queensland and improving the support, wellbeing, and retention of existing staff have been acknowledged by Queensland Health (p. 11) [25].

A project has been developed to implement and evaluate an innovative model that builds the capabilities of mid-level managers to perform their management responsibilities. The project empowers mid-level managers to better lead and support the health workforce at two HHSs in North Queensland. The proposed project is the first part of a larger body of work transitioning towards a new model that seeks to enhance management capacity. One of the anticipated outcomes include fostering partnership and collaboration amongst senior managers, clinician researchers, academic researchers at partner organisations, and researchers at James Cook University. The strengthened partnership and consolidation of management capacity will lay the groundwork for future-orientated capacity building in the organisations. The project will also capture learnings and evaluate the study design and strategies adopted in project implementation, laying the foundation for expanding the project beyond Northern Queensland.

The project will answer the following six questions:What are the difficulties mid-level managers face in fulfilling their management responsibilities?What approaches have mid-level managers implemented to support frontline managers/care staff in quality care provision? Have such approaches been effective? If not, why?What are the management strengths and areas for the development of the mid-level managers of partner organisations?What are the key factors that impact mid-level managers’ ability to support frontline managers/care staff in quality care provision?What strategies can be developed to tackle the management challenges faced by and the best support and empowerment of mid-level managers?To what extent are the project engagement strategies effective in formulating strategies to develop the management capabilities of mid-level managers in THHS and NSHHS?

## 2. Materials and Methods

### 2.1. Participating Organisations

Townsville Hospital and Health Service (THHS) and North West Hospital and Health Services (NWHHSs) are the participating health services that provide representation of regional, rural, and remote hospitals. In addition, there is strong organisational support for this project from the Executive level.

### 2.2. Team Formation and Project Co-Design

A team was formed with the required expertise from James Cook University, and senior managers and clinicians, and allied health professionals and mid-level managers from THHS and NWHHS. The formation of a team with representatives from key stakeholders is a fundamental co-design step to ensure the perspectives of key stakeholders are listened to throughout the project design, implementation, and evaluation process. In particular, the views of mid-level managers and key personnel from THHS and NWHHS were incorporated from the planning stage and throughout the project. A co-design approach was chosen to ensure that the project aim, research questions, and design generate evidence to help to solve the identified problems of both organisations. Co-design necessitates shared decision making and power amongst the participants and is driven by the need to address a problem [26]. As one of the key co-design principles, all team members will be actively engaged and continue working as part of the team throughout the project. It is anticipated, with organisational support, that the study participants are identified and the strategies developed in this research are acceptable and feasible to implement and be sustained in the clinical context of the partner organisations.

### 2.3. Research Design and Conceptual Consideration

Guided by a situation analysis approach, the project adopted a mixed-method multiphase design, including four key data collection steps. The project will first collect qualitative data reflecting the participants’ professional and lived experiences and preference for “interventions” that enable capability development. The findings will be integrated into two sets of questionnaires to further seek input from participants to understand various aspects of their experience in the workplace using six validated tools. A qualitative process will then be implemented to further seek the participants’ input in formulating “interventions”—the actual strategies that build the managers’ capability and management capacity of the organisation. Figure 2 details the project implementation stages, and the details of each stage are included in Table 1.

Through a quantitative process, before the construction of an innovative research model, two elements are required: (1) a thorough understanding of the current mid-level managers’ capabilities and the challenges in their working/operating environment and support, and (2) engaging in dialogue with mid-level managers and frontline managers/frontline staff to triangulate insights that will enable the construction of plans to further develop their capabilities. This will occur alongside the identification of strategies and supports that will enable the managers to best support frontline managers and care staff under their supervision [13].

### 2.4. Target Population

Primary target population: Mid-level managers who hold a Band 6 or 7 management positions from nursing, allied health/health promotion, and medical services. Based on the reporting structure as defined in Liang et al. (2013) [7], these positions are equivalent to the Level IV management level in large size hospitals or Level III management level in smaller sized hospitals and are responsible for the day-to-day operation of the organisation. There are approximately 182 and 170 mid-level managers currently employed at THHS and NWHHS, respectively.

Secondary target population: selected senior managers who supervise mid-level managers (for the focus groups) and frontline managers/frontline staff who are managed by mid-level managers (for both focus groups and online survey).

### 2.5. Methods and Steps for Data Collection

The project will be implemented via five stages. Details of each stage, including methods for data collection, participants, and estimated numbers, are included in Table 1.

### 2.6. Survey Instruments

For the managers’ online survey, the MCAP survey instrument, including the Management competency assessment tool * (MCAP Tool) [27,28] developed and validated in Australia and other countries [11,29], will be adapted. The survey includes:Selected management competency behavioural items from the MCAP Tool.Spritzer Psychological empowerment [30].Turnover Intention Inventory Scale [31].Management challenges.Preferred support and mechanism in management competency development and management capacity building.

The frontline managers and staff’ online survey includes:Job Satisfaction Index [32].Turnover Intention Inventory Scale [31].High-performing work system [33].Leader behavioural scale [33].Perception and preferred support from managers (informed by findings of the FGD).

Participants for both online survey will be asked to choose from the following three disciplines that best represent their work unit: medical, nursing, and allied health.

### 2.7. Participant Recruitment Strategies

For FGDs:

The Director of each organisation will email the FGD invitation and Participant Information Sheet to potential participants, who will be asked to contact the Principal Investigator should they like to participate in the FGD. Once contacted, the Principal Investigator will email the consent form for potential FGD attendees to complete and set up an FDG date suitable for all attendees.

For online surveys:

The Director of each organisation will send email invitations and the Participant Information Sheet to their senior and mid-level managers and frontline managers and staff, who will be asked to click on a link to the online survey set up at qualtics.com should they want to participate in the online survey. Implied consent is included on the front page of the survey.

### 2.8. Data Analysis

Qualitative data: The FGDs will be recorded, with the participants’ consent and a verbatim transcription approach. The thematic analysis of the transcripts will be guided by Braun and Clarke [34]. Selected members of the investigator team will become immersed in the data by listening, reading, and re-reading the transcripts. Checks for inter-coder reliability will be undertaken at various junctions during the data immersion and transcript-coding process. Microsoft Excel will be used for the thematic analysis. In the data analysis stage, the themes and subthemes will be sorted and ranked.

Quantitative data: Data collected from the online surveys will be downloaded from the Qualtrics website into Microsoft Excel. Following error checking, all data will then be imported into IBM SPSS Statistics version 25.0 (IBM Corp., Armonk, NY, USA) for analysis. The descriptive analysis and inferential analysis will be undertaken, including chi-squared, *t*-tests, and univariate analyses of variance. The alpha for all analyses will be set at 0.05. Pearson correlation coefficient may be performed to test the correlation between psychological empowerment or existing organisational support or the participation of management training and development opportunities and the management competency of mid-level managers.

### 2.9. Ethical Approval

Ethical approval was granted by Townsville Hospital and Health Services and James Cook University for conducting the research at both THHS and NWHHS (HREC/204/QTHS/106577). The preparation for hosting the first sets of focus group discussions has just commenced.

## 3. Discussion

Guided by the conceptual framework as detailed in Figure 1, the project has developed an innovative model of implementing a co-designed and active engagement process with close collaboration across institutions. The project brings together researchers, managers, and practitioners from different functional areas of the participating organisations, adding in-depth understanding of the roles and challenges both managers and care staff face. Drawing upon the lived experience, different backgrounds, and expertise of the research team will add gravitas and enhance the ability to reach the study population.

In recent years, co-design has been approved as an effective approach in developing implementation research and the formulation of improvement strategies and interventions. The use of co-design and active engagement principles to guide the research and project implementation process with participants will not only ensure evidence-based strategies that are relevant and sensitive to the institutional context can be developed [35,36], but also enhance the relevance of the research outcomes for mid-level managers, frontline staff, and ultimately, the healthcare consumers in North Queensland. The use of co-design principles engaging mid-level managers in identifying and developing solutions to problems they are currently experiencing is one mechanism towards realising the potential of the real-world impact of this research. Mounting evidence has proven how context can play a key influencing role in the selection of interventions and intervention outcomes [35,36,37]. At the organisational level, culture, strategic visions, governance structure, specific processes, and team composition are all influencing factors that must be taken into consideration during the strategy formulation process [37,38]. Only via a process of actively engaging individuals who may be affected by the strategies and interventions, all these factors can be fully considered.

The adoption of a multi-method and multiphase project design is the reflection of such understanding. The qualitative approach adopted as the first step of data collection allows the project and project team to draw on the live experience of participants and to develop an in-depth understanding of the issues they face [39] (managers and frontline staff). It can also inform the finalisation of the survey questionnaire that will be used during the quantitative data collection phase capturing the experience and views of the broader management and clinical teams. The project has been designed and will be implemented in two partner organisations in regional and rural Queensland with multiple service locations across regional, rural, and remote locations. Running multiple focus groups with both of the face-to-face and online options will maximise participation, ensuring the views and experiences of managers and frontline staff at the remote locations are also captured.

Hospital mid-level managers with dual clinical and management roles are key to quality patient care. Developing the mechanisms in supporting mid-level managers is a worthwhile investment for hospitals that not only improves managers’ capability in demonstrating their leadership and management competency in their roles, but also contributes to improved staff retention and job satisfaction [1,2,3,4]. In addition, developing managers’ leadership and management competencies should also be a key part of the human resource management policy. Without understanding the current support mechanisms of individual hospital, what works well and what does not, and what the actual needs of mid-level managers are, targeted and effective strategies may not be developed. The current project design, as illustrated in Figure 2, will allow strategies be developed and later implemented in two partner organisations, which will build the organisations’ capacity in providing better support to their managers, resulting in improved capabilities to lead and support frontline staff. Such improvements will ultimately increase staff job satisfaction and retention [1,5]. The active engagement of participants in co-designing interventions is essential, as they are more likely to lead to the research’s translation into practice [27,40].

Healthcare organisations operate in a resource-constrained environment; learning from past experiences and adopting tested tools, methods, and best practices in other sectors and countries are important and preferred. As part of this project, by adapting well-tested tools such as the MCAP tool [27,28], Spritzer Psychological Empowment [30], Job Satisfaction Index [32], Turnover Intention Inventory Scale [31], and High-performing work system [33] in the local organisation context, the study will develop contextually relevant strategies tailored to the needs and different support available at each partner organisation. The adoption of the tested tools is cost-effective for use in this novel study. It also improves the reliability and validity of the research data collection, resulting in comparable results being generated. Moreover, the project in turn will develop and test a new survey instrument by adopting the validated MCAP tool [27,28], which can capture the strengths of and identify areas for development in mid-level managers in leading and supporting frontline managers and staff. The MCAP tool can also identify deficiencies in mid-level managers’ support and the factors that build the management capabilities of healthcare organisations. Hence, a larger project will investigate how to build mid-level management capabilities at both the system and organisation levels for the health sector, leading to improved staff retention and job satisfaction and patient experience of care.

The study will generate evidence to guide the two partner organisations and similar organisations to develop strategies to better support mid-level managers. In addition, this study will build mid-level managers’ capabilities to support and lead frontline managers and staff, which is critical for high-quality patient care and to improve the outcomes of the population that they serve. A proportion of middle-level managers and care staff at NWHHS are Aboriginal and Torres Strait Islander peoples; so, the project hopes to develop a better understanding of their professional development needs, which ultimately leads to developing the workforce capacity to serve Aboriginal and Torres Strait Islander communities. In addition, this innovative project consisting of four stages with clearly defined outcomes provides a useful guide for healthcare organisations and researchers in building the management capacity of healthcare organisations using an evidence-based approach in the future.

This project brings together a team from academic institutions and hospital and health services with diverse experiences and expertise in research and practice. The diverse and complementary skill sets will be essential in knowledge transfer outcomes that enhance and increase research capacity and capabilities across the institutions. This study will be participatory in nature, and care will be taken to ensure the methods are culturally appropriate to regional, rural, remote, and Aboriginal and/or Torres Strait Islander populations across Northern Queensland.

## 4. Conclusions

This study protocol outlines an evidence-based approach to identify the competency development and support needs of mid-level managers and strategies in best supporting mid-level managers in their roles, which will contribute to retention and improved job satisfaction amongst frontline managers and frontline staff. This will ultimately benefit the quality of patient care provision in Northern Queensland. This project will utilise a mixed-methods multiphase design to capture rich data from managers at different management levels and frontline staff about the mid-level managers’ competency development and support needs. The identification of mid-level managers’ competency strengths and gaps will elucidate the development of context-specific strategies to best develop and empower mid-level managers. The use of co-design principles and a validated management competency tool, alongside the participation of two Hospital and Health Services and the health system and research expertise of the project team, are distinguishing factors that will assist in achieving the project goals.

## Figures and Tables

**Figure 1 ijerph-21-00994-f001:**
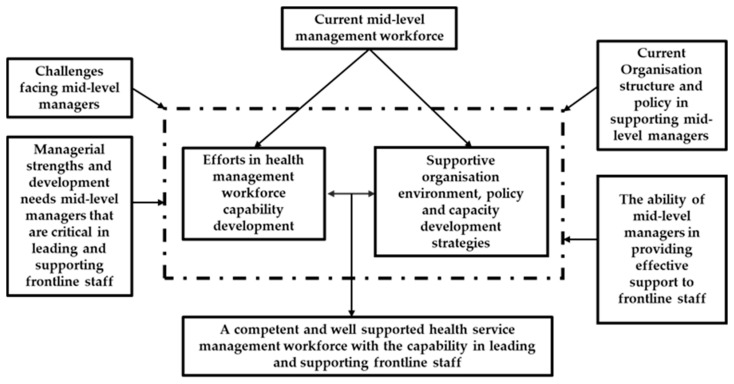
Conceptual framework for management development.

**Figure 2 ijerph-21-00994-f002:**
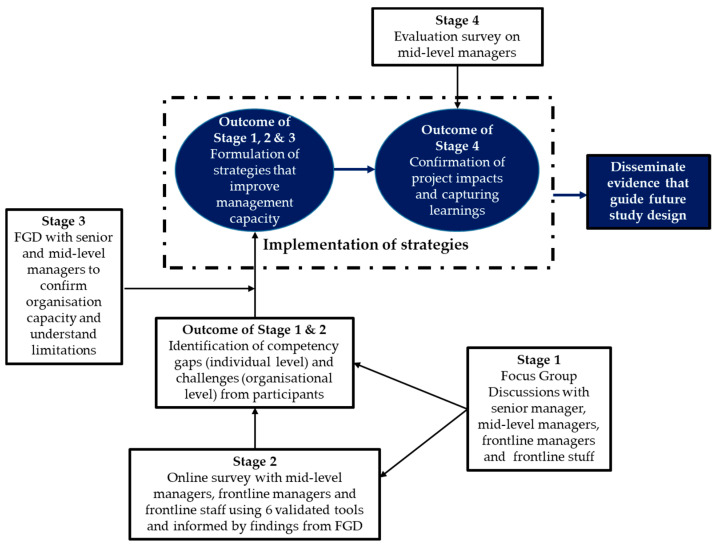
Detailed project overview: mixed-methods multiphase study.

**Table 1 ijerph-21-00994-t001:** Key steps for data collection.

Stages	Method for Data Collection	Participants	Sample Size	Notes
1	Two in-person focus group discussions (FGDs) at THHS and NWHHS, separately	Senior managers	6–8 for each FGD	To inform the revision of the survey instruments and partially address research questions (RQs) 1, 2, and 4
	* Two online FGDs for THHS and NWHHS, separately			
	Two in-person FGDs at THHS and NWHHS, separately	Mid-level managers	6–8 for each FGD	
	Two online FGDs for THHS and NWHHS, separately			
	Two in-person FGDs at THHS and NWHHS, separately	Frontline managers and frontline staff	6 frontline managers and 6 frontline staff for each FGD	
	Two online FGDs for THHS and NWHHS, separately			
2	Two online surveys	Mid-level managers	All will be invited. Minimum number is 65 × 2	Address RQs 1–4
		Frontline managers	130 × 2	
		Frontline staff	260 × 2	
3	Two in-person FGDs at THHS and NWHHS, separately	Senior and mid-level managers	6 senior managers and 6 mid-level managers for each FGD	Address RQ 5
	Two online FGDs for THHS and NWHHS, separately			
4	Online survey—evaluation	Mid-level managers	All will be invited. Minimum number is 65 × 2	Address RQ 6

* Online FGDs accommodate participants who work at remote locations.

## Data Availability

Not applicable as data are yet to be collected.

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
