# Peer review of "Empowering and Building the Capabilities of Mid-Level Health Service Managers to Lead and Support the Health Workforce—A Study Protocol"

_ijerph, 2024, doi:10.3390/ijerph21080994_

Round 1

Reviewer 1 Report

Comments and Suggestions for Authors

This is an interesting study.  There are certain areas to be clarified.  Co-design is usually for planning and development.  It can also be useful for research design. However, the managers should also be well familarised with the research process.  The authors can describe more on the process of co-design.

Situation analysis is mentioned but cannot see the result of analysis.  The analysis is to help design of study and not so clear how?

Needs justification of sample size for different qualitative survey and describe more on analysis of quantitative data.

Comments on the Quality of English Language

Minor proof read otherwise, fine.

Author Response

Dear Reviewer, 

Thank you very much for reviewing the manuscript and providing valuable feedback and suggestions. 

Please refer to to our responses to Reviewer 1 in the attached doc. 

Regards, 

Zhanming Liang on behalf of the co-authors. 

Reviewer 2 Report

Comments and Suggestions for Authors

This is a clear and concise description of a research protocol for a possibly interesting and useful study. I have just some comments and questions on some details:

- Given that the study is about mid-level managers, why only two focus groups per organization, while there are also two focus groups for senior managers?

- Given that the survey will be online, why is the sample size for mid-level managers only 65? Why not the all those managers? Otherwise, how will the sample be selected?

- Will it be possible to link survey responses of frontline managers and staff to those of the mid-level manager who is responsible for them?

- section 2.7 Why does the invitations for the FGD's and the survy go through the Director? Why not directly from the research team? It seems to me that the Director / Senior Managers have rather strong control over the study.

- Will the answers of the survy be anonymous to the researchers?

Author Response

Dear Reviewer, 

Thank you very much for reviewing the manuscript and providing valuable feedback and suggestions. 

Please refer to to our responses to Reviewer 2 in the attached. 

Regards, 

Zhanming Liang on behalf of the co-authors. 

Reviewer 3 Report

Comments and Suggestions for Authors

This is more like a research proposal which I don't think warrants publication.  There is nothing that unique here (according to me).  The study protocol is quite normal and can be used by any research...not only mid-level health managers.

Author Response

Dear Reviewer, 

Thank you very much for reviewing the manuscript and providing valuable feedback and suggestions. 

Please refer to to our responses to Reviewer 3 in the attached. 

Regards, 

Zhanming Liang on behalf of the co-authors. 

Round 2

Reviewer 3 Report

Comments and Suggestions for Authors

Thank you for your revisions.  My issue with the paper was different and did not warrant revisions so I will defer to the journal for the final decision.  Good luck with the study.

Author Response

Dear reviewer and editors,

No additional comments have been received after the 1st revision was submitted. We were invited to submit the 2nd revision by ensuring the manuscript meeting the 4000 words requirement.

We have added richer content throughout the manuscript, in particularly in the discussion section. Trust that the revised version will meet the publication requirements. 

Thank you for the opportunities

Regards, 

Zhanming Liang, on behalf of all authors. 
